# Generation of TRIM28 Knockout K562 Cells by CRISPR/Cas9 Genome Editing and Characterization of TRIM28-Regulated Gene Expression in Cell Proliferation and Hemoglobin Beta Subunits

**DOI:** 10.3390/ijms23126839

**Published:** 2022-06-20

**Authors:** Yao-Jen Chang, Zhifu Kang, Jiayuan Bei, Shu-Jen Chou, Mei-Yeh Jade Lu, Yu-Lun Su, Sheng-Wei Lin, Hsin-Hui Wang, Steven Lin, Ching-Jin Chang

**Affiliations:** 1Institute of Biological Chemistry, Academia Sinica, Taipei 11529, Taiwan; ntugeorge@gmail.com (Y.-J.C.); sanway@gate.sinica.edu.tw (S.-W.L.); 2Graduate Institute of Biochemical Sciences, College of Life Science, National Taiwan University, Taipei 10617, Taiwan; tomlichking@gmail.com (Z.K.); beijiayuan@gmail.com (J.B.); f94b46010@gmail.com (Y.-L.S.); 3Institute of Plant and Microbial Biology, Academia Sinica, Taipei 11529, Taiwan; sjchou@gate.sinica.edu.tw; 4Biodiversity Research Center, Academia Sinica, Taipei 11529, Taiwan; meiyehlu@gate.sinica.edu.tw; 5Department of Pediatrics, Division of Pediatric Immunology and Nephrology, Taipei Veterans General Hospital, Taipei 11217, Taiwan; hhwang@vghtpe.gov.tw; 6Department of Pediatrics, School of Medicine, College of Medicine, National Yang Ming Chiao Tung University, Taipei 112304, Taiwan; 7Institute of Emergency and Critical Care Medicine, College of Medicine, National Yang Ming Chiao Tung University, Taipei 112304, Taiwan

**Keywords:** TRIM28, CRISPR/Cas9, ddPCR, K562, hemoglobin beta, MAGEC2, SOX6, HBE

## Abstract

TRIM28 is a scaffold protein that interacts with DNA-binding proteins and recruits corepressor complexes to cause gene silencing. TRIM28 contributes to physiological functions such as cell growth and differentiation. In the chronic myeloid leukemia cell line K562, we edited *TRIM28* using CRISPR/Cas9 technology, and the complete and partial knockout (KO) cell clones were obtained and confirmed using quantitative droplet digital PCR (ddPCR) technology. The amplicon sequencing demonstrated no off-target effects in our gene editing experiments. The *TRIM28* KO cells grew slowly and appeared red, seeming to have a tendency towards erythroid differentiation. To understand how TRIM28 controls K562 cell proliferation and differentiation, transcriptome profiling analysis was performed in wild-type and KO cells to identify TRIM28-regulated genes. Some of the RNAs that encode the proteins regulating the cell cycle were increased (such as p21) or decreased (such as cyclin D2) in *TRIM28* KO cell clones; a tumor marker, the MAGE (melanoma antigen) family, which is involved in cell proliferation was reduced. Moreover, we found that knockout of *TRIM28* can induce miR-874 expression to downregulate *MAGEC2* mRNA via post-transcriptional regulation. The embryonic epsilon-globin gene was significantly increased in *TRIM28* KO cell clones through the downregulation of transcription repressor SOX6. Taken together, we provide evidence to demonstrate the regulatory network of TRIM28-mediated cell growth and erythroid differentiation in K562 leukemia cells.

## 1. Introduction

TRIM28 (tripartite motif-containing protein 28), also known as KAP1 (KRAB domain-associated protein 1) and TIF1β (transcriptional intermediated factor 1β), belongs to the tripartite motif family, which contains an N-terminal ring finger, a B box, and a coiled-coil leucine zipper (RBCC) domain. The RBCC is necessary for interaction with a Krüppel-associated box (KRAB)-containing zinc-finger proteins (ZFPs) to silence genes and form oligomers [1,2,3,4]. The mechanism of TRIM28-mediated gene repression involves the recruitment of HDAC (histone deacetylase) complex NuRD and histone H3 lysine 9-specific methylatransferase SETDB1 by both the C-terminal PHD and the bromodomain [5,6]. The central PxVxL pentapeptide associates with the heterochromatin protein 1 (HP1) [7]. KRAB-ZFP and TRIM28 complex-mediated chromatin DNA recognition is accompanied by DNA methylation [8]. These studies support the role of TRIM28 in the epigenetic regulation of transcription via the maintenance of heterochromatin structures. In addition, the RING domain of TRIM proteins serves as an E3 ubiquitin ligase [9]. The MAGE (melanoma antigen) family proteins assemble with RING E3 ligase and function as the substrate adaptors that enhance ubiquitination activity [10,11]. MAGE family proteins are overexpressed in many cancers and lead to tumor growth and metastasis [12].

TRIM28 is a crucial regulator in development and differentiation. Knockout of the *Trim28* gene in mice results in embryonic lethality, indicating that it plays an essential role in embryonic development [13]. It is required for the maintenance and pluripotency of embryonic stem cells [14,15] and for controlling the gene regulatory network in hematopoietic stem cell development, including T-cells, B-cells, and erythropoiesis [16,17,18,19,20]. Recently, a report showed that the knockdown of TRIM28 increases K562 cell differentiation [21].

Human erythroleukemia K562 cells were derived from the pleural effusion of a patient with chronic myelogenous leukemia (CML) in terminal blast crisis [22]. They contain a Philadelphia chromosome with a BCR-ABL fusion oncogene [23]. K562 is a model to study cellular differentiation and can undergo further differentiation into megakaryocytic or erythroid lineages depending on the stimuli [24]. During erythroid differentiation, hemoglobin gene transcription is initiated at the erythroblast stage. The beta globin gene cluster undergoes the sequential expression of the embryo epsilon, then the fetal gamma, and then the adult delta and beta globin genes [25]. It has been shown that K562 cells usually induce the expression of epsilon- and gamma-globin (HBE and HBG) [26]. Transcription factors such as SOX6 can bind to the promoter of *HBE* and *HBG* to inhibit their expression [27,28,29]. Furthermore, TRIM28 may associate with TR2/TR4 orphan nuclear receptors to repress *HBE* and *HBG* gene expression [30].

To investigate the functional roles of TRIM28 in leukemia, we knocked out TRIM28 in K562 cells using CRISPR/Cas9 genome editing. The partial and complete KO clones were confirmed using amplicon sequencing and droplet digital PCR (ddPCR). The phenotype and gene expression analyses indicate that TRIM28 affects cell proliferation and differentiation into erythroid cells. The detailed gene regulatory network was further studied, and we found that knockout of *TRIM28* induced *HBE* gene expression via the downregulation of transcription repressor SOX6 and decreased cell growth-related MAGEC2 by miR-874 activation.

## 2. Results

### 2.1. Knockout of TRIM28 in K562 Using CRISPR/Cas9 Mediated Genome Editing

To explore the functional roles of TRIM28 in hematopoietic cells, we performed CRISPR/Cas9 genome-editing in K562 cells. Two double-strand break sites on *TRIM28* at once were generated using the nucleofection of a pair of Cas9-RNPs with sgRNAs targeting two different genomic loci of *TRIM28*, which can introduce a large deletion on the *TRIM28* gene after non-homologous end joining (NHEJ) DNA repair (Figure 1a). After a preliminary test, the combination of Cas9-RNP with *TRIM28*-sgRNA1 and 3 for 48 h incubation after nucleofection showed the best editing efficiency and was used for the following *TRIM28* knockout (Figure 1a and Appendix A). The edited cells were sorted into single cells for culture, and the genotype of each clone was determined by genomic PCR. Four clones, 4C, 9E, 6E, and 10D, were collected (Figure 1b). Compared with the WT K562 cells, the PCR products of clones 4C and 9E contained an upper band close to 1500 bp, the expected 900 bp lower band as a KO allele, and a middle band, which may be from a single cut deletion within either targeting site; however, we only observed the lower band in 6E and 10D clones. The aberrant triploid karyotype in K562 cells has been demonstrated [31], and there are three copies of *TRIM28* in the genome. We assumed that clones 4C and 9E were possibly partial KO and that 6E and 10D were complete KO due to full editing within both targeting sites. The protein expression levels of TRIM28 were significantly decreased in 4C and 9E clones and showed no expression in 6E and 10D clones (Figure 1c).

We further examined the integrity of CRISPR by amplicon sequencing [32]. We followed the method published by Sedlazeck and his colleagues [33]. Using PacBio Sequel we performed the amplicon sequencing of the on-target site and the top three off-target sites for sgRNA1 and sgRNA3 in our four KO clones with PacBio long-read sequencing, showing very diversified DNA structure variance on *TRIM28* genomic loci (Appendix A). Importantly, these top three off-target sites had no off-target editing detected in these four *TRIM28* KO clones. A quantitative droplet digital PCR (ddPCR) was performed to determine the ratio of *TRIM28* knockout alleles as described in [34,35,36]. Based on the signal intensity ratio of target to reference, ddPCR clearly revealed the structural variation in each KO clone (Figure 2). Based on the results of amplicon templates deep sequencing and quantitative ddPCR, we demonstrated that the clone of 9E is a partial KO with only one intact *TRIM28* allele and two KO alleles; and 6E is a complete KO clone. They were used for further characterization of phenotype and gene expression.

### 2.2. Knockout of TRIM28 Inhibits the Cell Proliferation

To investigate whether the TRIM28 level in K562 cells is functionally linked to the malignant characteristics of fast proliferation in CML blast crisis cells, we measure the cell proliferation in WT and *TRIM28* KO K562 clones. As shown in Figure 3a, the copy number associated expression level of TRIM28 was directly correlated with the cell duplication rate between the WT and the *TRIM28* partial- and complete-KO clones. Usually, the repression of cell proliferation is caused by arresting a specific part of the cell cycle. Accordingly, FACS analysis of PI-stained cells revealed that the portion of cells in the G0/G1 phase was increased by approximately 10% (WT: 31.2%, partial-KO: 42.1%, complete-KO: 42.3%) and the proportion of cells in the S phase decreased by a comparable degree (WT: 44.6%, partial-KO: 32.8%, complete-KO: 34.5%) (Figure 3b). The cellular proliferation was also estimated by direct measurement of DNA synthesis via the incorporation of EdU (5-ethynyl-2′-deoxyuridine) (Figure 3c). When comparing the portion of S phase cells and the EdU+ cells, the number of EdU+ cells was approximately 1.4 times higher in WT cells than in KO cells (Figure 3c). These independent data indicate that the reduction in TRIM28 decelerates K562 proliferation.

### 2.3. TRIM28 Regulates Cell Proliferation-Related Gene Expression

To explore the molecular mechanism of TRIM28-mediated cell proliferation in K562 cells a, we used the Affymetrix Clariom D assay, Human microarray platform to reveal the change in gene expression of the transcriptome profiling in WT, *TRIM28* partial- and complete-KO K562 cells. The results showed a TRIM28 dosage-dependent change of several genes, which were demonstrated to be involved in the cell cycle progression, especially in BCR-ABL CML (e.g., *CDKN1A* (p21) and *CCND2* (Cyclin D2)), which all responded to the G1/S phase progression (Figure 4a). Additionally, the MAGEA, B, and C protein families comprise class I MAGEs, a group of highly homologous proteins whose expression is suppressed in all normal tissues except developing sperm [37], and function as drivers of tumorigenesis [38]. As shown in Figure 4b, the mRNA expression levels of *MAGEA9*, *B1*, and *C2* were downregulated in *TRIM28* KO cells depending on the expression levels of TRIM28. Their gene expression levels were verified by RT (reverse transcription)-qPCR (quantitative PCR) (Figure 4c). The results were consistent with the microarray analysis, indicating that TRIM28 might play functions in cell proliferation via controlling related gene expression.

### 2.4. TRIM28 Involves in MAGEC2 mRNA Expression

MAGEC2 promotes proliferation and resistance to apoptosis in multiple myeloma [39]. We are interested in how MAGEC2 expression is regulated in K562 leukemia cells. First, we verified the effect of TRIM28 in MAGEC2 expression. We overexpressed FLAG-Trim28 in *TRIM28* complete-KO K562 cells. After drug selection, RNAs and proteins were isolated for the RT-qPCR and Western blotting analyses. As shown in Figure 5a, the RNA and protein levels of MAGEC2 were increased when ectopic expression of Trim28 (Figure 5a). The previous report demonstrated that MAGEC2 protein stability is dependent on the interaction with TRIM28 [40]. We transfected the FLAG-MAGEC2 expression vector into HEK293T and performed IP-Western blotting. As shown in Figure 5b, FLAG-MAGEC2 can precipitate the endogenous TRIM28. However, the MAGEC2 protein levels were not restored in the *TRIM28*-KO K562 cells treated with proteasome inhibitor MG132 (Figure 5c). These indicate that TRIM28 modulates MAGEC2 expression via affecting RNA expression, not protein stabilization.

### 2.5. TRIM28 KO Induces miR-874 to Downregulate MAGEC2 Expression

We want to understand how *TRIM28*-KO results in the downregulation of *MAGEC2* mRNA. From our ChIP-qPCR result, TRIM28 did not directly target the *MAGEC2* promoter (data not shown). We propose that TRIM28 indirectly decreases *MAGEC2* via miRNAs upregulation. We found the *MAGEC2* 3′UTR contains the miR-874-targeted sequence from a Targetscan survey and a previous report [41] (Figure 6a). The miR-874 was upregulated in *TRIM28* complete-KO cells, and restoration of TRIM28 expression in complete-KO cells would decrease miR-874 expression (Figure 6b). The chromatin-IP showed that TRIM28 was enriched on the *miR*-*874* promoter (Figure 6c). Ectopic expression of miR-874 downregulated *MAGEC2* 3′UTR-mediated luciferase activity in 293T cells (Figure 6d) and also decreased *MAGEC2* RNA and protein expression in WT K562 cells (Figure 6e). Together, our results suggest knockout of *TRIM28* decreases *MAGEC2* gene expression via increasing miR-874 levels.

### 2.6. TRIM28 Regulates HBE mRNA Expression via SOX6 Transcription Repressor

Since K562 cells can be induced to carry out erythroid differentiation, we would like to know whether TRIM28 regulates this process. We observed that *TRIM28* complete-KO cells were reddish like erythrocytes. The microarray analysis showed that the embryo *HBE* was highly increased in *TRIM28* complete-KO cells (Figure 7b). We verified the hemoglobin beta gene cluster expression including *HBE* and *HBG* by RT-qPCR (Figure 7a). The results showed that *HBE* and *HBG* were significantly upregulated in *TRIM28* complete-KO cells and slightly increased in *TRIM28* partial-KO cells. We want to identify the specific target transcription factors involved in TRIM28-mediated beta-globin expression. Previous reports demonstrated that transcription factor SOX6 repressed mouse embryonic epsilon globin expression [27]. Moreover, in our microarray results, the downregulated *SOX6* mRNA was correlated with the upregulation of *HBE* mRNA in *TRIM28* complete-KO cells (Figure 7b). RT-qPCR verification showed *SOX6* mRNA was decreased in *TRIM28* complete-KO K562 cells and was increased when TRIM28 overexpression (Appendix A). When overexpression of SOX6 or TRIM28 occurred in *TRIM28* complete-KO cells, the expression level of *HBE* was dramatically decreased (Figure 7c). The co-IP demonstrated the interaction between TRIM28 and SOX6 (Figure 7d). The ChIP assay showed that both TRIM28 and SOX6 were enriched on the *HBE* promoter (Figure 7e). When SOX6 was knock down by lentivirus-carrying shRNA, the TRIM28 enrichment on the *HBE* promoter was decreased and *HBE* mRNA expression was increased (Appendix A). The results indicate that TRIM28 targets the *HBE* gene, leading to a decrease in *HBE* gene expression through SOX6. Ectopic expression of SOX6 and TRIM28 in 293T cells resulted in the downregulation of *HBE* promoter (containing two possible SOX6-binding sites)-driven luciferase activity (Figure 7f). This suggests that TRIM28-SOX6 is a repressive complex for *HBE* gene expression and that TRIM28 is involved in SOX6 gene expression.

## 3. Discussion

This study designs a strategy to generate and examine the *TRIM28* gene knockout by CRISPR/Cas9 (Figure 1). We use the short-lived Cas9-RNP mediated editing method instead of nucleic acid-based delivery of Cas9 to reduce the off-target editing rate [42]. Two double-strand break sites on *TRIM28* were generated to introduce a large deletion for the follow-up single clone genomic PCR screening. After clone isolation, we prepared an amplicon to perform PacBio long-read sequencing and confirm that there were no off-targets (Appendix A). Finally, the copy number variation of the genome structure was demonstrated by quantitative ddPCR, and the *TRIM28* partial-KO and complete-KO clones were obtained (Figure 2).

The leukemia K562 cell has been used extensively as a model of a common progenitor for the study of stem cell differentiation and as a platform for drug treatment tests. We knocked out *TRIM28* in K562 cells, resulting in cell proliferation inhibition (Figure 3). The expression level of cell cycle-regulated gene *CDKN1A* was increased and *CCND2* was decreased in *TRIM28*-KO cells, correlating with proliferation inhibition (Figure 4). The cell growth-related MAGE family expression was reduced in *TRIM28*-KO K562 cells (Figure 4). This indicates that TRIM28 may be involved in the tumorigenesis of leukemia. The multi-domain of TRIM28 contributes to the regulation of a variety of cellular processes such as cell proliferation, apoptosis, protein degradation, autophagy and EMT, and several reports have shown the positive clinical relevance between TRIM28 expression levels and specific cancer types [43]. Compared to normal samples, *TRIM28* mRNA level is higher in blood-related cancer patients and cell lines [21,44].

We observed that both *HBE* and *HBG* were induced in *TRIM28*-KO K562 cells (Figure 7), indicating that the cells were in the process of erythrocyte differentiation. During the development of human erythrocytes, there is a switch from the expression of the embryonic *epsilon*–*globin* gene to the fetal *gamma*–*globin* gene in utero, and the *gamma*–*globin* gene is silenced postpartum as *beta*–*globin* gene expression becomes predominant. SOX6 and orphan nuclear receptor TR2/TR4 (NR2C1/NR2C2) heterodimer were involved in silencing *HBE* expression [27,28,30]. Consistent with the increased expression of *HBE* in *TRIM28* complete-KO K562 cells, the *SOX6* expression was reduced in the KO cells (Figure 7). We showed that TRIM28 interacts with SOX6 and both SOX6 and TRIM28 bind to the *HBE* promoter (Figure 7). On the other hand, TRIM28 not only interacts with SOX6 protein to repress *HBE* expression but also modulates *SOX6* gene expression. TRIM28 did not target the *SOX6* promoter directly in the ChIP assay (data not shown). Some of our data suggest that *TRIM28* KO induces miR-18a, which targets *SOX6* mRNA 3′UTR, leading to the downregulation of SOX6 mRNA (Appendix A). Since TRIM28 can recruit histone and DNA modification enzymes for gene silencing [45], it may be involved in the molecular mechanism of the HDAC and DNA methylation inhibitor-mediated erythrocyte differentiation of k562 cells.

The MAGE family consists of about 60 genes defined by a common conserved MAGE homology domain (MHD), including type I: MAGE-A, -B, and -C and type II: MAGE-D, -E, -F, -G, -H, -L2 and Necdin families, serving as cancer biomarkers and immunotherapeutic targets due to their specific expression pattern in various cancers [38,46]. They are also called cancer–testis antigens (CTA) due to their expression being restricted to the testis and being aberrantly re-expressed in cancers. MAGEs are associated with poor clinical prognosis and function as drivers of tumorigenesis [38]. For example, MAGEC2 promotes proliferation and resistance to apoptosis in multiple myeloma [39]. The MAGE family proteins assemble with RING E3 ligase and function as substrate adaptors that enhance ubiquitination activity [10,11]. The substrate-binding cleft (SBC) had been identified in MAGEA11 and the mutation of SBC can disrupt its oncogenic activity [11]. It has been found that MAGEA2, -A3, -A6, and -C2 interact with TRIM28 [47]. TRIM28-MAGE complex promotes p53, KRAB-ZFPs, AMPK, and FBP1 (fructose-1,6-bisphosphatase) ubiquitination and degradation [48,49,50,51]. Due to the known relationship between TRIM28 and MAGEC2, we would like to know how TRIM28 regulates MAGEC2 expression (Figure 5). It was known that DNA methylation, histone modification, and non-coding RNAs were involved in the gene expression of MAGEs in cancers [12]. In addition, it has been reported that the miR-874 could directly bind to 3′UTR of MAGEC2 and destabilize RNA in cancer cells [41]. Because TRIM28 is an epigenetic corepressor, its depletion could cause gene activation. Thus, we suggest that TRIM28 regulates MAGEC2 expression indirectly via miR874-mediated mRNA degradation (Figure 6). miR-874 is downregulated to negatively regulate several mRNAs in several cancers and has been suggested to be a tumor suppressor gene [52,53,54,55]. These regulatory axes demonstrate that TRIM28 plays a function in cancer cell proliferation and metabolism.

## 4. Materials and Methods

### 4.1. Cell Lines and Cell Culture

The human CML cell line K562 (ATCC, Cat# CCL243) was maintained in Roswell Park Memorial Institute medium (RPMI 1640) (Hyclone Laboratories, Logan, UT, USA), supplemented with 100 U/mL of penicillin, 100 μg/mL of streptomycin (Hyclone), 2 mM glutamine (Hyclone), and 10% fetal bovine serum (Gibco, Grand Island, NY, USA) in a humidified atmosphere at 37 °C in a 5% CO_2_ incubator. Human embryonic kidney (HEK) 293T (ATCC; Cat# CRL-3216) cells were maintained in Dulbecco’s modified Eagle medium (DMEM) (Hyclone) supplemented with 10% FBS (Gibco), 100 U/mL of penicillin, and 100 μg/mL of streptomycin (Hyclone) at 37 °C in a 5% CO2-humidified atmosphere.

### 4.2. Plasmids

The pcDNA3.1-HA-TRIM28 expression plasmid was synthesized by GenScript (Piscataway, NJ, USA). The FLAG-Trim28 expression plasmid was constructed as described in [56]. The *MAGEC2* 3′UTR-containing luciferase reporter was constructed by subcloning the PCR product of 3′UTR using the primers: forward AGTCTAGGATAGTTTCTTCC and reverse CAGTGGGGCTTTCTTTTATTAAG into 3′ of pCMV-Tag2C-Luc (Stratagene, San Diego, CA, USA) and sequences confirmed. The *HBE* promoter was PCR cloned using the primers: forward CAAGCCAGAAGGAGGAACTG and reverse GCTCCTCATCTATCTGCAAC and K562 genomic DNA as templates. The PCR product was sequence confirmed and ligated into pGL3-basic (Promega, Madison, WI, USA) for luciferase assay. The Myc-DDK-SOX6, and -MAGEC2, and miR-874 expression vectors were purchased from OriGene (Rockville, MD, USA).

### 4.3. CRISPR/Cas9-Mediated TRIM28 Knockout in K562 Cells

*TRIM28* knockout was performed by nucleofection of a Cas9 ribonucleoprotein (RNP). Cas9 sgRNAs targeting *TRIM28* were designed using the CRISPR Design Tool at www.benchling.com (accessed on 19 September 2016). Only sgRNAs with high predicted off-target scores (more precise) were selected (Appendix A). The sgRNAs were synthesized by in vitro transcription and purified by denaturing PAGE as described [57]. Each purified sgRNA was refolded into a functional structure in buffer containing 20 mM HEPES pH 7.5, 150 mM KCl, 10% glycerol, 1 mM 2-mercaptoethanol, and 1 mM MgCl_2_. Cas9 RNP was prepared by incubating purified recombinant Cas9 and sgRNA at 1:1.2 molar ratio at 37 °C for 10 min. Nucleofection of human CML K562 cells was performed in a Lonza 4D Nucleofector system using an SE Cell Line 4D-Nucleofector^TM^ kit and an FF-120 pulse (Lonza, Basel, Switzerland). After nucleofection, the cells were incubated at 37 °C for 48 h and single cells were sorted by a FACSJazz automated cell sorter (BD, Bergen County, NJ, USA).

### 4.4. Analysis of TRIM28 Knockout by Genomic PCR

Genomic DNA was extracted using QuickExtract™ DNA Extraction Solution (Epicentre Biotech, Madison, WI, USA). Cells were mixed into 200 µL of QuickExtract Solution and vortexed for 15 s. The tubes were incubated at 65 °C for 6 min. After 15 s of vortexing, samples were incubated at 98 °C for 2 min. The DNA was stored at −20 °C or −80 °C for long-term storage according to the manufacturer’s protocol. The 1490 bp region of TRIM28, containing the Cas9-RNP targeting site, was PCR amplified using the TRIM28 gDNA primer set (forward 5′-CTCTAC ATCTTCCCA ATA AATGGCCCAGTG-3′, and reverse 5′-TGTGAACAAAGCAGAACCCTCTGCCTCAGT-3′). A 590 bp fragment was deleted between exon 5 and exon 8 on the *TRIM28* gene in knockout cells; thus, the size of the PCR product generated by this primer pair in knockout mutants will be reduced to 900 bp.

### 4.5. Copy Number Variation Detection by Quantitative Droplet Digital PCR (ddPCR)

We used 50 ng of total genomic DNA from WT K562 Cell or each TRIM28-KO clone to determine the copy number of functional *TRIM28* in the genome by the ddPCR method. The ddPCR was performed using a QX200™ Droplet Digital™ PCR System (Bio-Rad, Hercules, CA, USA) with QX200™ ddPCR™ EvaGreen^®^ Supermix (Bio-Rad). Four probe and primer sets were designed, including ddPCR TRIM28_Left and ddPCR TRIM28_Right for the double-strand break site of TRIM28-sgRNA1 and TRIM28-sgRNA3, respectively, ddPCR TRIM28_Middle as the specific deletion region, and ddPCR TRIM28_Ref., a specific intact *TRIM28* gene, as a reference.

### 4.6. RNA Isolation, Reverse Transcription and Quantitative PCR

We harvested 5 × 10^6^ K562 cells and lysed them with 1 mL TRIzol reagent (Invitrogen, Carlsbad, CA, USA). The RNA was isolated according to the manufacturer’s protocol. After quantification by A 260/280 measurements, 2 μg RNA was treated with DNase I (Invitrogen) and then reverse transcribed into cDNA using Superscript IV (Invitrogen) following the manufacturer’s instructions. Quantitative PCR was performed with the Corbett Research RG-6000 Real-Time PCR Thermocycler (Qiagen, Germantown, MD, USA). The total volume was 20 μL including QuantiNova SYBR Green master mix (Qiagen), 20-fold diluted cDNA, and 0.3 μM forward and reverse primers as shown in Appendix A. The amplification conditions were 60 cycles at 95 °C for 10 s and 60 °C for 15 s. The results were analyzed by the 2^−∆∆Ct^ relative quantitation method. The miRNAs were isolated using a mirVana miRNA isolation kit (Invitrogen) and quantitated using a TaqMan miRNA assay (Thermo Fisher Scientific, Waltham, MA, USA).

### 4.7. Clariom D Whole Transcriptome Microarray Analysis

Total RNA was isolated from WT, TRIM28 partial-KO and complete-KO K562 cells using TRIzol reagent (Invitrogen). After integrity analysis using Agilent Bioanalyzer, the samples were processed to the Affymetrix gene expression service lab at Academia Sinica. A total of 300 ng of total RNA was used for cDNA synthesis, labeled by in vitro transcription followed by fragmentation according to the manufacturer’s protocol (GeneChip^TM^ Whole Transcript Expression Arrays, Affymetrix, Thermo Fisher Scientific). The labeled samples were hybridized to the Clariom D Assay Human Chip (Affymetrix, Thermo Fisher Scientific) at 45 °C for 16.5 h. After washing, the chip was stained by Fluidic Station-450 and scanned with Affymetrix GeneChip Scanner 3000. The raw data were QC and analyzed using Transcriptome Analysis Console (TAC) 4.0 (Affymetrix, Thermo Fisher Scientific).

### 4.8. Preparation of Whole Cell Extracts and Western Blot Analysis

For the immunoprecipitation assay, the cells were harvested by spinning for 5 min at 5000× *g*, and then washed with ice-cold PBS twice. Cell pellets were resuspended by gently pipetting in protease inhibitor completed cold whole-cell extraction buffer (20 mM HEPES (pH 7.6), 10% glycerol, 0.4 M NaCl, 0.5% Triton-X 100, 5 mM EDTA, 1 mM EGTA, 1mM dithiothreitol, 1mM PMSF, 1 mg/mL leupeptin, 1 mg/mL pepstatin A, and NaF) about 10 cell pellet volumes. The cell lysis was performed at 4 °C for 30 min with vigorous vortexing. After centrifugation for 15 min at 14,000 rpm at 4 °C, the total protein concentration of the supernatant was determined by Protein Assay Dye Reagent (Bio-Rad) for Western blotting analysis. The proteins were separated in SDS-PAGE and transferred to PVDF membrane (PerkinElmer, Waltham, MA, USA) in the transfer buffer (48 mM Tris-Cl, 39 mM glycine, 0.375% SDS, 20% methanol) for 1–1.5 h at a constant current (2.0 mA/cm^2^) by a semidried plate (Bio-Rad). After transfer, the membrane was blocked with a blocking solution (5% non-fat milk in PBST (phosphate-buffered saline containing 0.1% Tween-20)) for 30 min at room temperature. Western blot analysis was performed by incubating the membrane with appropriately diluted first antibodies in blocking solution at room temperature for 2 h or 4 °C overnight. After washing with PBST three times (10−15 min each), the membrane was incubated with diluted HRP-conjugated secondary antibodies in a blocking solution for 1 h at room temperature. Finally, the membrane was washed again with PBST three times for 15 min each. The immunocomplexes were visualized by enhanced chemiluminescence (PerkinElmer) and then subjected to x-Ray film (Fuji, Tokyo, Japan).

### 4.9. Cell Proliferation and Cell Viability Assay

WT, *TRIM28* partial-KO and complete-KO K562 cells were seeded in a 6 cm plate in 2 × 10^5^/mL. Cell proliferation was determined by LUNA™ Cell Counter with 0.4% trypan blue stain (Logos Biosystems, Gyeonggi-do, Korea). Values are derived from an average of three independent experiments at the indicated time points. Data were normalized to day 0 of 2 × 10^5^/mL and used Log_2_ fold change as cell duplication rate. A total of 5 × 10^3^ cells were plated in 96-well plates and cultured with the indicated compound for 48 h. After culturing, cell viability was measured using a CCK-8 kit (Enzo, New York, NY, USA). The percentage of cell viability was determined relative to the untreated controls. Experiments were repeated at least three times with triplicate samples.

### 4.10. Cell Cycle Analysis

The cell cycle profile was analyzed by Propidium Iodide Flow Cytometry Kit (Abcam, Cambridge, UK). Cells were harvested in single-cell suspension and washed with 1X PBS once, followed by fixing cells in 66.6% ice-cold ethanol overnight at 4 °C and staining with propidium iodide (PI) following the manufacturer’s instructions for use. The DNA content was analyzed by FACSCalibur™ Flow Cytometry (BD). The distribution of cells in the G0/G1, S, and G2/M phases was further analyzed by FlowJo (FlowJo, Ashland, OR, USA).

### 4.11. Immunoprecipitation Assay

Whole-cell extracts were pre-incubated with Protein-G-Sepharose beads (Sigma-Aldrich, St. Louis, MO, USA) for 1 h rotating at 4 °C to avoid non-specific binding during the immunoprecipitation process, and then the pre-cleared protein samples were incubated with the G-Sepharose beads conjugating desired antibodies for 2–3 h rotating at 4 °C. We briefly span down the Sepharose beads with the bound immunocomplexes (3000 rpm, 30 s) and subsequently washed them with ice-cold WCE buffer three times, finally boiling them in 4X SDS-sample buffer (95 °C, 10 min).

### 4.12. ChIP Assay

A total of 1 × 10^7^ leukemia cells were harvested and fixed with 1% formaldehyde and quenched with 125 mM glycine. Isolated chromatin was sonicated by a bioruptor (Diagenode, Denville, NJ, USA), and immunoprecipitated with antibodies and Dynabeads (Thermo Fisher Scientific) in buffer containing 16.25 mM Tris at pH 8.0, 137.5 mM NaCl, 1 mM EDTA, 0.5 mM EGTA, 1.25% Triton X-100, protease inhibitors) overnight. Antibodies against TRIM28 (homemade) or SOX6 (ABclonal, Wobum, MA, USA) or normal IgG were used to IP chromatin. After extensive washes, the ChIP samples were used for SYBR Green qPCR (Qiagen) using indicated primers (Appendix A) to perform triplicate experiments. The enrichment folds were presented by the relative occupancy calculated as a ratio of specific signal (from anti-TRIM28) over background (from anti-normal IgG). All experiments were independently repeated at least three times.

### 4.13. Overexpression of Trim28, SOX6 or miR-874 in TRIM28 Complete-KO K562 Cells

The FLAG-Trim28, FLAG-SOX6, miR-874, or vector was transfected into *TRIM28* complete-KO K562 cells using Lipofectamine LTX and Plus reagent (Invitrogen) following the manufacturer’s protocol. After selection with 800 μg/mL of G418 (Thermo Fisher Scientific) for three weeks, the surviving cells were harvested and RNA and protein were isolated for RT-qPCR and Western blotting analysis, respectively.

### 4.14. Dual Luciferase Reporter Assay

The 293T cells grown in 24-well plates were co-transfected with MAGEC2 3′UTR-containing luciferase reporter (125 ng per well), the miR-874 expression vector (200 ng per well), and 100 ng of Renilla using Turbofect (Invitrogen). Forty-eight hours later, a Dual-Luciferase Reporter Assay kit (Promega) was used to measure the luciferase and Renilla activities according to the manufacturer’s instructions. The relative luciferase activity was determined using BioTek Synergy 2 (BioTek, Winooski, VT, USA), and the transfection efficiency was normalized to Renilla activity.

### 4.15. Statistical Analysis

The results are presented as the mean ± confidence intervals (CI) of at least three independent experiments. The statistically significant values were calculated by one-way ANOVA with the post-hoc Tukey HSD test. The significance was labeled as * *p* < 0.05, ** *p* < 0.01, or ns (non-significant difference).

## 5. Conclusions

Leukemia is one of the major cancers that causes a high rate of mortality all over the world. Research and development on the identification of novel anti-leukemic targets are important for disease therapy. Since stalled differentiation is considered a characteristic of leukemia, promoting leukemia cell differentiation has become the focus of medical research. Our study shows that TRIM28 promotes cell proliferation and inhibits cell differentiation in leukemia cells as it is one anti-leukemic target. Furthermore, TRIM28 forms a functional network through miRNA repression (miR-874) and MAGEC2 induction. Therefore, the overexpression of miR-874 mimic or immuno-targeted MAGEC2 might be another approach to leukemia therapy. Based on these results, more TRIM28-regulated miRNA and MAGE family expression can be extensively explored to build up a complete interacting network in leukemia, which is beneficial for diagnosis, therapy, and prognosis.

## Figures and Tables

**Figure 1 ijms-23-06839-f001:**
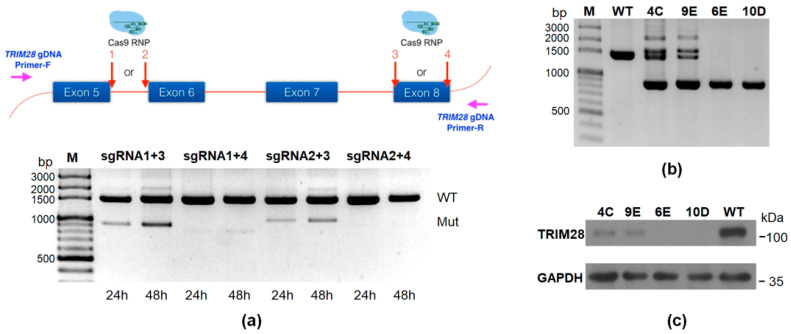
Knockout of *TRIM28* in K562 cells. (**a**) Strategy of *TRIM28* knockout in K562 cells. Two double-strand break sites on TRIM28 were generated at once using the nucleofection of a pair of Cas9-RNAs as indicated. After incubation for 24 h or 48 h, the editing efficiency was determined by PCR with *TRIM28* gDNA primer set; the upper bands around 1500 bp (labeled as WT) are considered as having no editing, and the lower bands indicate a deletion of genomic DNA between the two editing sites. (**b**) The single cell was separated by an automatic cell sorter. Four clones (4C, 9E, 6E and 10D) were isolated after genomic PCR. (**c**) TRIM28 protein expression analysis in these knockout clones.

**Figure 2 ijms-23-06839-f002:**
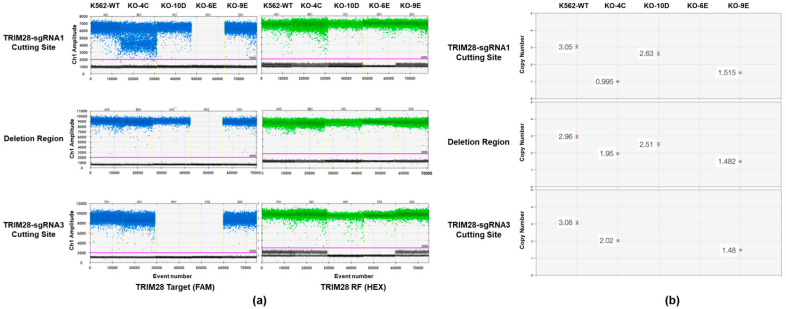
Confirmation of *TRIM28* knockout by ddPCR. (**a**) The horizontal axis indicates the event number of five reactions including wild-type (WT) and four *TRIM28* KO clones separated by yellow lines, and the vertical axis indicates sample amplitude. The left panel is performed by FAM-labeled TRIM28-target primers located at the sgRNA1 cutting site, deletion region, and sgRNA3 cutting site, respectively, and the right panel by HEX-labeled reference primers. The positive and negative droplets classified by thresholds (pink lines) of individual wells are shown in blue (or green) and grey, respectively. (**b**) The copy number of specific sites in each clone was determined by the copy number of reference times the ratio of target to reference (Target copy = Reference copy X (Target intensity/Reference Intensity)).

**Figure 3 ijms-23-06839-f003:**
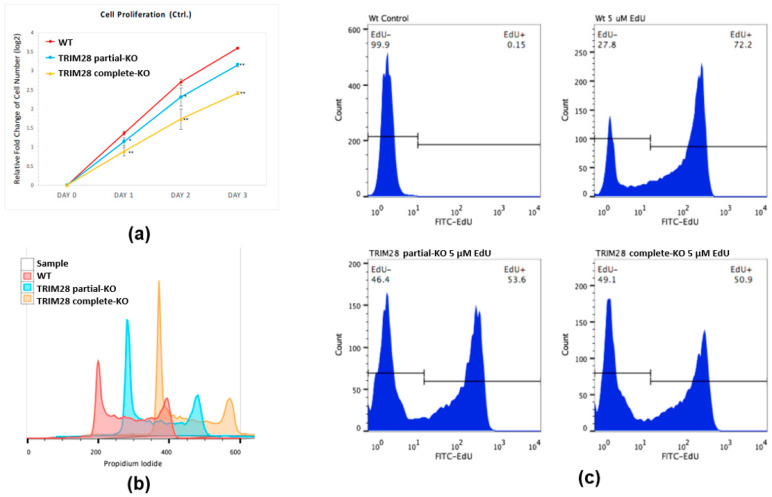
The expression level of TRIM28 regulates K562 cell proliferation. (**a**) The cell proliferation of WT, partial-KO and complete-KO K562 was determined by cell counting. Values are derived from an average of three independent experiments at the indicated time points. Data were normalized to day 0 (2 × 10^5^) and used Log2-fold change as cell duplication rate. * *p* < 0.05; ** *p* < 0.01 comparison to the WT K562 cells. (**b**) The cellular DNA content of WT, partial-KO, and complete-KO K562 was analyzed by flow cytometry. (**c**) EdU labeling showing proliferation of WT, partial-KO, and complete-KO cells.

**Figure 4 ijms-23-06839-f004:**
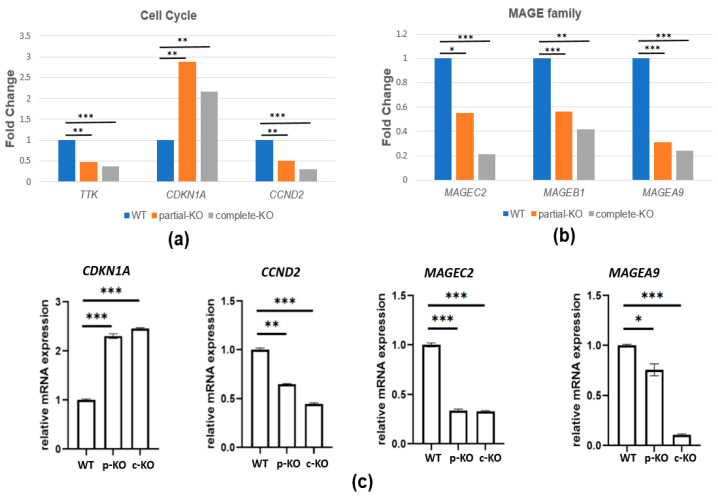
TRIM28 regulates cell proliferation-related gene expression. The transcripts from WT, *TRIM28* partial-KO (p-KO), and complete-KO (c-KO) K562 cells were performed in a human microarray Clariom D assay. (**a**,**b**) The expression levels of cell cycle and MAGE family genes are dependent on TRIM28 dosages. (**c**) RT-qPCR analysis of *CDKN1A*, *CCND2*, *MAGEC2*, and *MAGEA9* expression. * *p* < 0.05, ** *p* < 0.01, and *** *p* < 0.001.

**Figure 5 ijms-23-06839-f005:**
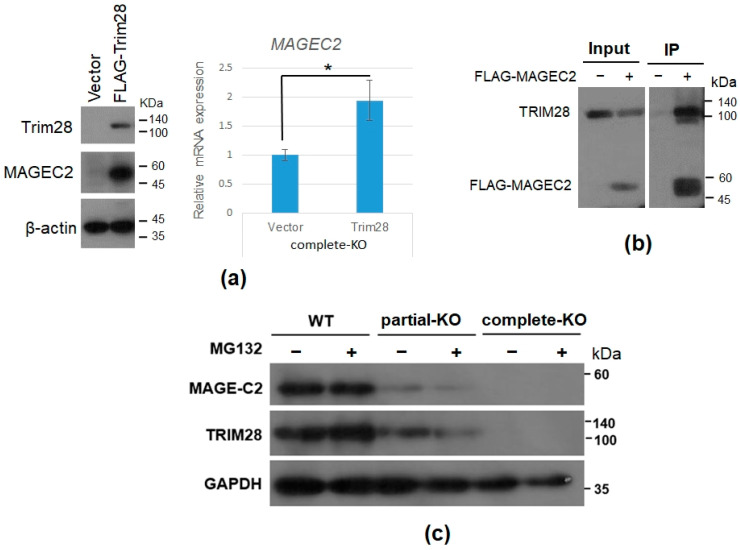
TRIM28 involves in *MAGEC2* mRNA expression. (**a**) Overexpression of Trim28 in *TRIM28* complete-KO K562 cells rescues MAGEC2 mRNA and protein expression. * *p* < 0.05. (**b**) Immunoprecipitation of FLAG-MAGEC2 with TRIM28 in HEK293T cells. (**c**) MAGEC2 protein expression under MG132 treatment. The WT, *TRIM28* partial-KO, and complete-KO K562 cells were treated with 20 μM of MG132 for 6 h and whole cell extracts were isolated for Western blotting analysis using indicated antibodies.

**Figure 6 ijms-23-06839-f006:**
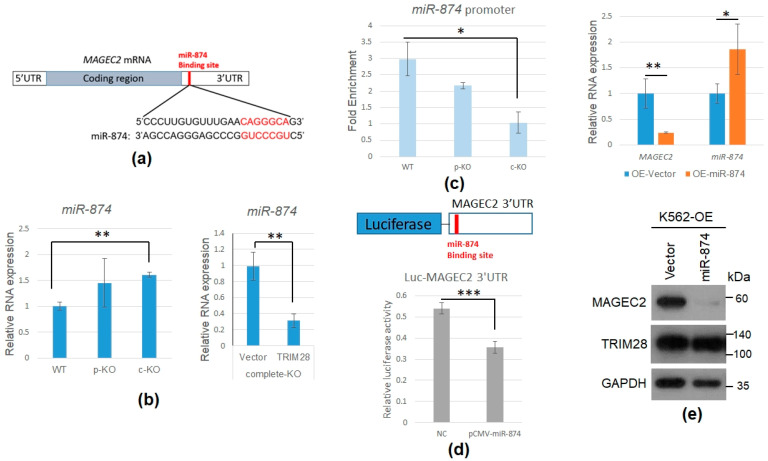
*TRIM28* KO induces miR-874 to downregulate *MAGEC2* expression. (**a**) *MAGEC2* mRNA 3′UTR contains a miR-874 targeted element. (**b**) miR-874 was induced in *TRIM28* complete-KO K562 cells (**left** panel). Overexpression of Trim28 in complete-KO K562 cells decreased miR-874 expression using qPCR analysis (**right** panel). (**c**) ChIP assay. The WT, partial-KO (p-KO), and complete-KO (c-KO) cells were fixed by formaldehyde and IPed with anti-TRIM28. The precipitated genomic DNA was qPCR analyzed with *miR-874* promoter primers. (**d**) Ectopic expression of miR-874 downregulates *MAGEC2* 3′UTR-containing luciferase reporter in HEK293T cells. (**e**) The WT K562 cells were overexpressed (OE) with a cloning vector or *miR-874* expression plasmid. After drug selection, RNAs were isolated for qPCR analyses with *MAGEC2* or *miR-874* primers (**upper** panel), and whole-cell extracts were Western blotted with indicated antibodies (**lower** panel). * *p* < 0.05, ** *p* < 0.01, and *** *p* < 0.001.

**Figure 7 ijms-23-06839-f007:**
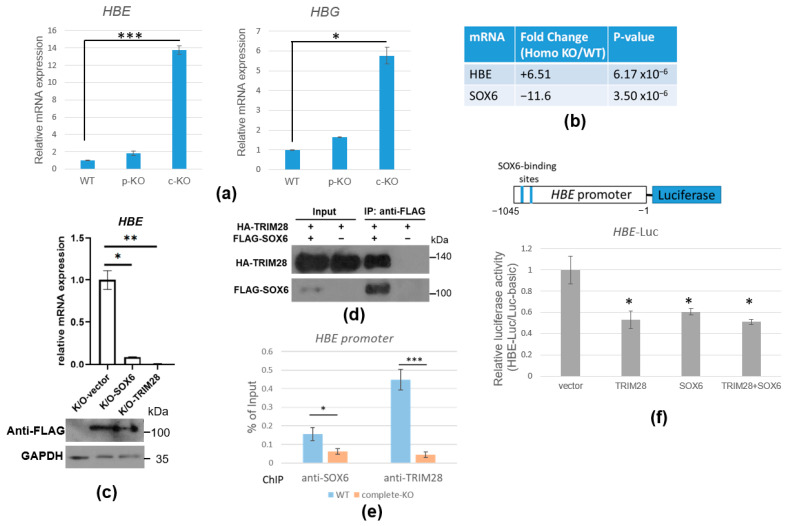
TRIM28 regulates *HBE* mRNA expression via SOX6 transcription repressor. (**a**) RT-qPCR analysis. RNAs from WT, *TRIM28* partial-KO, and complete-KO K562 cells were subjected to RT-qPCR with *HBE*, *HBG*, and *GAPDH* primers. (**b**) The *HBE* and *SOX6* mRNA expression in microarray analysis. (**c**) The rescue analysis. The *TRIM28* complete-KO K562 cells were transfected with FLAG-SOX6 or FLAG-Trim28. After drug selection, RNAs were isolated for RT-qPCR analyses with *HBE* cDNA primers (upper panel) and whole-cell extracts were Western blotted with indicated antibodies (lower panel). (**d**) Co-IP of TRIM28 and SOX6. The FLAG-SOX6 and HA-TRIM28 expression plasmids were transfected as indicated into HEK293T cells. The whole-cell extracts were IPed with anti-FLAG and Western blotting with anti-HA and anti-FLAG. (**e**) ChIP assay. The WT, partial-KO, and complete-KO cells were fixed by formaldehyde and IPed with anti-TRIM28 or anti-SOX6. The precipitated genomic DNA was qPCR analyzed with *HBE* promoter primers. (**f**) Luciferase assay. The *HBE* promoter-driven luciferase expression vector was co-transfected with FLAG-SOX6, HA-TRIM28, or FLAG-SOX6 and HA-TRIM28 together and then luciferase activity was measured. All experiments were independently repeated three times, and * *p* < 0.05, ** *p* < 0.01, and *** *p* < 0.001.

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
