# Peer review of "Generation of TRIM28 Knockout K562 Cells by CRISPR/Cas9 Genome Editing and Characterization of TRIM28-Regulated Gene Expression in Cell Proliferation and Hemoglobin Beta Subunits"

_ijms, 2022, doi:10.3390/ijms23126839_

Round 1
Reviewer 1 Report
This work provides a preliminary analysis of cell proliferation and gene expression following knockout of TRIM28 in K562 cells. The authors show that knockout of TRIM28 inhibits cell proliferation by differentially regulating cell cycle-related genes and downregulates the expression of MAGEC2 and globin genes through miR-874 and SOX6, respectively.
- My main concern is that both contents and writing of the manuscript were not organized in a logic manner. The data were assembled together without a common thread. In particular, there is apparently no connection between different aspects observed after TRIM28 knockout: reduced cell proliferation, decreased expression of MAGEC2, and increased expression of HBE. Substantial reorganization is necessary to make the manuscript more logic and meaningful.
- TRIM28 physically interacts with SOX6, but it is not clear whether TRIM28 binds to HBE promoter in a SOX6-dependent or -independent manner.
- In section 2.6 the authors claim that TRIM28 regulates HBE expression through SOX6, but this was not clearly demonstrated. TRIM28 regulates SOX6 mRNA expression, binds to HBE promoter and physically interacts with SOX6 protein. Therefore, TRIM28 could regulate HBE expression by binding to its promoter, through SOX6 mRNA expression or physical interaction with SOX6 protein. The authors should clarify these possibilities.
- Since TRIM28 regulates SOX6 mRNA expression, why the authors did not perform ChIP assay with SOX6 promoter? Therefore, it is not clear how TRIM28 regulates SOX6 expression.
- Since TRIM28 regulates MAGEC2 mRNA levels through miR-874 and figure 6e clearly shows miR-874 inhibits MAGEC2 mRNA expression to cause absence of the corresponding protein, I do not understand the logic of testing MAGEC2 protein stability as shown in figure 5c.
- Section 2.1, “uncleofection” should be “nucleofection”.
- Section 2.4, TRIM28 knockout downregulates MAGEC2 mRNA expression. The title is not appropriate because in this section the authors did not show downregulation of MAGEC2 mRNA expression in TRIM28 knockout cells. The data were presented in section 2.3.
- The authors should precise the four clones (4C, 9E, 6E, and 10D) are generated by which combination of sgRNAs.
- Error bars should be indicated in figure 4a and b.
- To claim that TRIM28 knockout downregulates MAGEC2 mRNA expression through miR-874, the authors need to block miR-874 in TRIM28 knockout cells using inhibitors.
- Figure 6 legend, cells were fixed with formamide?
- Figure 7f, statistical significance was not indicated.
Reviewer 2 Report
The article describes the targeted KO of TRIM28 gene in K562 cells and some characterization of the cellular phenotype.
While the overall study appears sound, there are some editorial issues with English Language usage as well as multiple grammatical errors.
Specific Concerns
The authors need to explain in the results that K562 are triploid in the region of the genome containing the TRIM8 gene at the onset of the results section. Consequently, the terms hemi-KO and homo-KO may not be appropriate as they are usually referring to a diploid genome so it may be more appropriate to use an alternative terminology for clarity (e.g. 2 KO alleles/1 WT allele) for hemi-KO.
The authors should include some text in the results explaining that single cell clones were isolated as the reader has to read the methods to determine what was done.
The authors need to clarify the statements about absence of off -target effects as the text as written implies that the entire genome was surveyed but rather was only three potential off target sites. They also mention a PAC bio sequencing but it is not clear in the results what was sequenced used this approach – the TRIM8 gene – but from where to where?
To indicate that the K562 have differentiated may be inaccurate as fully differentiated K562 tend to stop proliferating entirely. It would perhaps be more appropriate to simply state that the KO cells are expression globin genes.
Minor Points
There are multiple mis-spelling and errors throughout the manuscript.
Two examples early in the manuscript.
Line 83 - using uncleofection of
Line 90 - an upper band closed to 1500 bp
Please go through the manuscript with a spell checker and correct the spelling errors (not just the few listed above)
There are also multiple instances of non-standard English that will need to be corrected.
e.g Line 88
We monitored the genome type of each single…
Not sure what the authors mean here? Are they referring to the TRIM28 alleles?
Figure 1 legend
are consider as no editing, and the lower bands (Mut) show editing within both site of each combination
Should be
Are considered as having no editing, and the lower bands indicate a deletion of genomic DNA between the two editing sites.
These are just two examples, the entire manuscript needs to be edited for non-standard English.
Round 2
Reviewer 1 Report
The authors have made substantial efforts to revise the manuscript. They also performed a few additional experiments in response to my comments. The revised manuscript shows signs of improvement but there are still important issues that need to be clarified.
1. The function of Sox6 in regulating globin gene expression and terminal differentiation of red blood cells have been reported previously, using K562 cells (for example PMID: 21263153). The present results need to be discussed with respect to those previous works.
2. In figure S3, the efficiency of Sox6 knockdown by shRNA seems to be rather low. I would like to see how Sox6 protein level was affected. In the legend, (d) should be (b).
3. CRISPR/Cas9 should be written consistently.
4. The manuscript needs proof editing, particularly the titles of different sections. For example, "TRIM28 involves in MAGEC2 RNA expression".
